# Evaluating Bargaining Skills in Online Second-Hand Marketplace with LLM Seller Agents

## Abstract

In online second-hand marketplaces, multi-turn bargaining is a crucial part of seller-buyer interactions. Large Language Models (LLMs) can act as *seller agents*, negotiating with buyers on behalf of sellers under given business constraints. A critical ability for such agents is to track and accurately interpret cumulative buyer intents across long negotiations, which directly impacts bargaining effectiveness. We introduce a multi-turn evaluation framework for measuring the bargaining ability of seller agents in e-commerce dialogues. The framework tests whether an agent can extract and track buyer intents. Our contributions are: (1) a large-scale e-commerce bargaining benchmark spanning 622 categories, 9,892 products, and 3,014 tasks; (2) a turn-level evaluation framework grounded in Theory of Mind (ToM), enabling detailed assessment of model performance beyond outcome-only metrics; and (3) an automated pipeline that constructs intent annotations and evaluation data from large-scale dialogues, transferable across datasets and negotiation domains.

## 1 Introduction

Bargaining is a fundamental social intelligence skill with substantial economic impact across industries. In e-commerce, bargaining is equally critical: studies on negotiation tasks such as Craigslist-Bargain He et al. (2018) and more recent applied systems like FishBargain Dexin and Xu (2025) illustrate that effective bargaining improves user experience, increases platform conversion rates, and ultimately drives revenue. For AI agents aiming to make real impact in commercial contexts, mastering bargaining is therefore a necessary step.

From an intelligence perspective, bargaining sits at the intersection of natural language understanding, strategic reasoning Qian et al. (2025), and Theory of Mind (ToM) modeling. It requires interpreting scenario-specific information, reasoning about counterpart goals and constraints Davidson et al. (2024). Effective bargaining further demands tracking buyer intents across multiple turns, consistently recalling past commitments, and applying such understanding under domain-specific constraints Dexin and Xu (2025)—capabilities where current Large Language Models (LLMs) remain fragile.

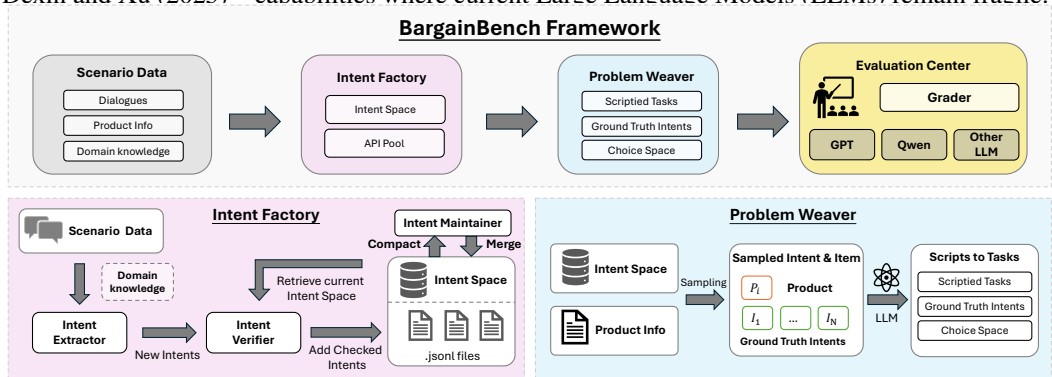

Figure 1: BargainBench framework: **Intent Factory** extracts an intent space, **Problem Weaver** generates scripted dialogues, and **Evaluation Center** scores LLM performance.

Current LLMs face distinct challenges in bargaining beyond those seen in standard dialogue tasks. First, *intent drift*, where buyer goals evolve subtly across turns, forces models to maintain a coherent belief state while detecting implicit shifts in strategy. Second, *contextual memory degradation*, meaning models forget earlier commitments or constraints once dialogues exceed typical context windows, leading to inconsistent responses. Third, *adversarial misalignment*, when buyers deliberately exploit ambiguity or use deceptive tactics, exposes gaps in models trained mainly on cooperative dialogue. These vulnerabilities are amplified in commercial settings, where business rules, compliance requirements, and reputation management add further complexity.

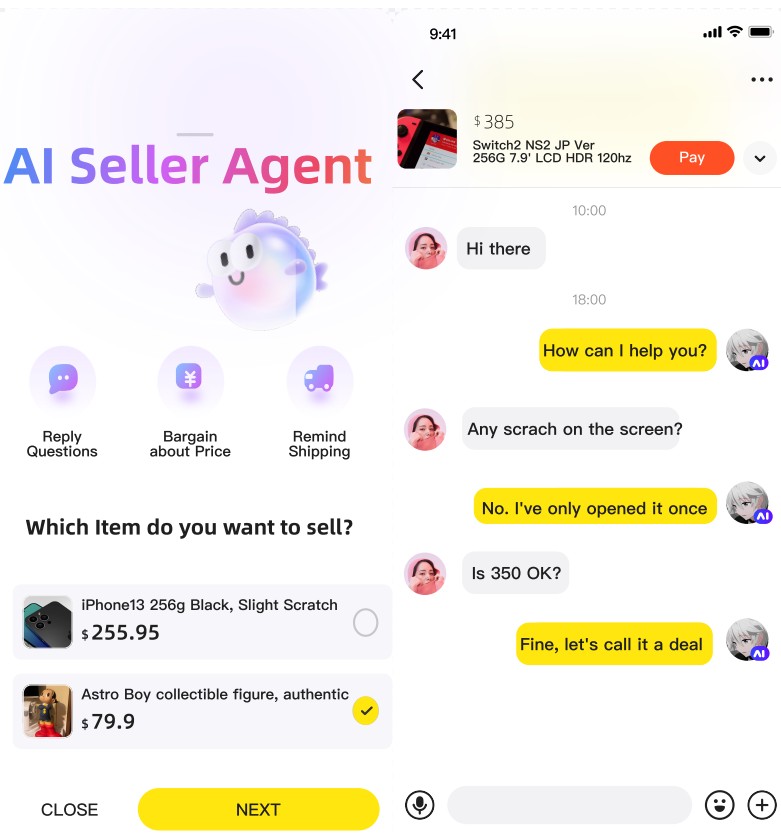

Figure 2: Left: Human sellers select items and delegate them to the AI seller agent. Right: Human buyers interact with the AI agent to ask questions and negotiate prices.

To address these challenges, we propose grounding bargaining evaluation in Theory of Mind (ToM) principles Chan et al. (2024). Rather than evaluating only final negotiation outcomes—which conflate strategic success with intent understanding—our approach assesses whether models can accurately infer and track the mental states, intentions, and constraints of their negotiation partners at each conversational turn. This shift from outcome-based to process-based evaluation enables in-depth diagnosis of model capabilities, identifies specific failure modes, and provides actionable insights for targeted improvement.

Prior benchmarks either ignore real-world constraints or score only final deals Xia et al. (2024), missing the intermediate reasoning processes that shape negotiation success. Existing approaches often simplify bargaining to basic offer-counteroffer exchanges, overlooking the rich information-seeking and the exploration of hard constraints such as price limits, product condition, or return policies that characterize realistic negotiations. We introduce the seller-agent setting, where agents negotiate on behalf of sellers under explicit business rules and product constraints, to isolate and measure turn-level intent understanding in a controlled yet realistic environment.

Our framework extends beyond e-commerce applications to provide a general methodology for evaluating intent understanding in multi-turn, goal-oriented dialogues. The core *intent–action–tool* hierarchy offers a domain-agnostic structure that can be systematically adapted to diplomatic negoti-

ations, medical consultations, educational tutoring, legal mediation, and other scenarios requiring sustained reasoning about counterpart mental states. This generalization capability positions our work as a foundational contribution to the broader study of social intelligence evaluation Tang et al. (2025), with direct implications for designing human–AI interaction across diverse domains.

Our work makes four main contributions. (1) We present **BargainBench**, a large-scale bargaining benchmark covering 622 product categories, 9,892 listings, and 3,014 evaluation tasks, with authentic business constraints and real-world complexity. (2) We propose a **turn-level evaluation framework** grounded in Theory of Mind, providing ground-truth buyer intents and shifting the focus from outcome-only metrics to reasoning processes that support sustained negotiation. (3) We design an **automated pipeline** for extracting high-quality intent annotations from large-scale dialogue data, enabling reproducible and scalable benchmarking while preserving annotation consistency. (4) We highlight the **cross-domain potential** of our approach: the intent–action–tool hierarchy and turn-level protocol are broadly applicable and can be adapted to settings such as diplomatic negotiations, medical consultations. This positions BargainBench as a foundation for future studies on universal intent-understanding evaluation.

## 2 RELATED WORK

**Multi-turn negotiation benchmarks and tasks.** Early datasets such as DealOrNoDeal Lewis et al. (2017) and CraigslistBargain He et al. (2018) established text-based bargaining protocols by modeling buyer–seller negotiation as multi-turn dialogues with real listed products. Later work has moved toward more interactive and applied domains. Xia et al. Xia et al. (2024) formalize bargaining as an asymmetric incomplete information game, while Davidson et al. Davidson et al. (2024) evaluate model agency through negotiation games with both self-play and cross-play settings. More applied systems, such as FishBargain Dexin and Xu (2025) and debt collection negotiation frameworks Wang et al. (2025), extend bargaining research to real-world domains with vertical application requirements Zhu et al. (2025). Despite these advances, most benchmarks continue to emphasize final outcomes such as success rate or profit, leaving the intermediate reasoning processes that drive negotiation effectiveness underexplored.

**Intent recognition and tracking in dialogue.** Dialogue State Tracking benchmarks such as MultiWOZ Budzianowski et al. (2018) address explicit slot filling and task goals, but bargaining typically involves *implicit, evolving, and context-dependent intents*. NegotiationToM Chan et al. (2024) introduces belief and intention modeling for negotiation dialogues, showing that even advanced LLMs struggle with consistent inference across turns. Guan et al. Guan et al. (2025) survey methods for multi-turn conversation evaluation and highlight intent tracking as a key challenge. More broadly, Theory-of-Mind (ToM) research investigates whether LLMs demonstrate human-like mental state reasoning. Kosinski Kosinski (2024) reports positive performance on classical false-belief tasks, emphasizing the need for more robust benchmarks. Our setting narrows these discussions to negotiation-specific buyer intent extraction and turn-level tracking.

**Tool-augmented agents and process-grounded evaluation.** Tool-augmented benchmarks have stressed robustness and correctness under domain constraints. $\tau$-Bench Yao et al. (2024) evaluates tool–agent–user interaction in rule-constrained environments. ToolACE Liu et al. (2025) builds large-scale function-calling datasets through synthetic generation, while ACEBench Chen et al. (2025) categorizes tool-use evaluation into multiple multi-agent and ambiguous scenarios. These lines of work highlight the importance of aligning user intent, action selection Ye et al. (2025), and tool execution. In our setting, bargaining agents must ground buyer intents into seller-side actions (e.g., price adjustment, shipping changes, proof requests), providing a natural case study of the *intent–action–tool* hierarchy.

**Comparison to Prior Work.** Most existing benchmarks assess bargaining only by final outcomes such as success rate or payoff, without capturing how agents reason across turns Lewis et al. (2017); He et al. (2018); Xia et al. (2024); Davidson et al. (2024). Applied systems including FishBargain Dexin and Xu (2025) and debt-collection negotiations Wang et al. (2025) also primarily report transaction completion or recovery metrics, reflecting the same emphasis on end results. What remains missing is an evaluation that explicitly examines the intermediate reasoning process—whether models can track

shifting goals, remember prior commitments, and align actions with task constraints—capabilities essential for realistic negotiations but underexplored in outcome-only evaluations. In contrast, our benchmark adopts a data-oriented task generation method, enabling not only turn-level evaluation in bargaining but also straightforward adaptation of the same methodology to other multi-turn, goal-driven dialogue domains.

| Name | Scalability | Ground Truth | # Tasks | Avg. Turns | # Items | # Categories | Avg. Description Length | Evaluation |
|---|---|---|---|---|---|---|---|---|
| CraigslistBargain (Lewis et al., 2017) | ✗ | ✗ | 6,682 | 9.2 | 1,402 | 6 | 31 words | Human eval+Outcome Metrics |
| DealOrNoDeal (He et al., 2018) | ✗ | ✗ | 5,808 | 6.6 | 3 | 20 | ✗ | Outcome Metrics |
| Measuring Bargaining Abilities (Xia et al., 2024) | ✓ | ✗ | 930 | N/A | 930 | 18 | Short texts | Outcome Metrics |
| **BargainBench (ours)** | **✓** | **✓** | **3,014** | **3** | **9,892** | **85** | **55 words** | **Turn-level Intents ✓** |

Table 1: Comparison of bargaining benchmarks in E-commerce field. BargainBench uniquely provides scalability and intermediate ground-truth annotations, along with broader product coverage, more listed items and longer descriptions.

## 3 METHODOLOGY

To evaluate bargaining ability under realistic yet verifiable conditions, we design **BargainBench**, a three-stage pipeline consisting of the **Intent Factory**, **Problem Weaver**, and **Evaluation Center** (Figure 1). Unlike outcome-based negotiation benchmarks, our framework isolates the capability of *understanding* bargaining context: each turn is paired with explicit ground-truth intent, enabling reproducible and interpretable evaluation. This design exploits the asymmetry between authoring and solving—models can readily generate convincing multi-turn dialogues when given target intents, yet often fail to recover those intents from completed exchanges. By preserving verifiable ground truth at each turn, BargainBench delivers interpretable and transferable performance measurements, and can be readily adapted to other goal-oriented domains such as diplomatic negotiations and cooperative games.

At the core of the framework is a hierarchical decomposition of negotiation behavior into three levels:

- *Intent* — the most abstract, high-level buyer goals
- *Action* — mid-level strategies that operationalize an intent
- *Tool* — the most atomic, directly verifiable move expressed in a single utterance

This structure balances abstraction with verifiability: intents capture long-horizon goals, tools ensure evaluation can be grounded in unambiguous turn-level labels, and actions provide the bridge that groups semantically similar tools into coherent strategies. The hierarchy not only disentangles complex dialogues but also supports multi-granularity diagnosis of model strengths and weaknesses.

The three modules operationalize this design: the **Intent Factory** constructs and refines the hierarchy from raw dialogues, the **Problem Weaver** instantiates bargaining tasks grounded in real-world product metadata, and the **Evaluation Center** executes controlled turn-level testing with standardized metrics.

### 3.1 INTENT FACTORY

The **Intent Factory** distils raw dialogues, product data, and domain knowledge into the three-level *intent–action–tool* hierarchy and a compact API pool. To ensure both coverage and consistency, we employ a lightweight multi-agent pipeline: The **Extractor** identifies candidate intent–action–tool triplets from both dialogues and product metadata. The **Verifier** checks whether these extracted intents are valid, discarding duplicates or entries already present in the intent space. The **Expert_Guide** leverages domain and expert knowledge to provide a cold-start foundation, re-labeling ambiguous cases and guiding the construction of a coherent intent hierarchy. Finally, the **Maintainer** clusters semantically similar entries, merges redundancy, and outputs a compact but comprehensive hierarchy.

We assess the quality of the mined hierarchy with two metrics: *Coverage* (the proportion of ground-truth intents recalled) and *Duplicate Ratio* (the proportion of redundant entries remaining). Detailed definitions and formulas are provided in the Appendix A. In practice, Coverage remains consistently above 95%, so the main effect of the pipeline is to progressively reduce redundancy, producing a more compact and semantically consistent intent space that is better suited for large-scale task generation.

The resulting output of this process is the finalized **Intent Space**, which serves as the foundation for subsequent task construction and evaluation.

## 3.2 PROBLEM WEAVER

The **Problem Weaver** turns abstract entries from the Intent Space into concrete, multi-turn bargaining tasks grounded in real product metadata (title, description, price, category). For each selected product, it first samples a *ground-truth intent* as the answer key, checks plausibility under product constraints, and then generates turn-specific *Buyer Messages*. Each turn is paired with (i) a system prompt and (ii) an *intent choice space* (a candidate set that includes the ground truth and distractors). The process repeats until the target number of turns is reached, yielding an independent, evaluation-ready task object (dialogue script, per-turn labels, and choice sets). A task example appears in Figure 10; prompt templates are given in Appendix B.

**Inputs**

- Product set $\mathcal{P}$ (real item metadata)
- Intent pool $\mathcal{I}$ (from Intent Factory)
- Prompt templates $\mathcal{T}$ (for Buyer Messages)
- Target turns $L_{\text{target}}$; choice size $|\mathcal{C}|$ (e.g., 20)

**Outputs**

- Task set $\mathcal{D}$ of multi-turn dialogues
- Per-turn ground-truth intents and choice spaces
- Evaluation-ready JSON objects (script + labels + metadata)

---

**Algorithm 1:** Problem Weaver Pipeline

**Data:** Products $\mathcal{P}$,;
Intents $\mathcal{I}$,;
Templates $\mathcal{T}$,;
$L_{\text{target}}, |\mathcal{C}|$
**Result:** Task set $\mathcal{D}$
**foreach** *product* $p \in \mathcal{P}$ **do**
    Sample ground-truth intent $i^* \in \mathcal{I}$;
    **if** $i^*$ *is plausible for* $p$ *(consistent with product attributes and rules)* **then**
        Initialize empty dialogue $d$;
        **while** $|d| < L_{target}$ **do**
            Generate Buyer Message using $(p, i^*, \mathcal{T})$;
            Build choice space $\mathcal{C} \subset \mathcal{I}$ with $i^*$ + distractors, $|\mathcal{C}|$ fixed;
            Attach system prompt and ground-truth label $i^*$; append turn to $d$;
        Package $d$ as a JSON task (script, per-turn labels, choice spaces);;
        add to $\mathcal{D}$;

---

## 3.3 EVALUATION CENTER

The **Evaluation Center** executes these scenarios on target LLMs and scores predictions against turn-level ground truth, with the maximum per-dialogue score equal to the number of annotated intents. The evaluation checks whether outputs meet format requirements, predicted intents exist in the intent space, and the predicted sequence matches the reference. At each turn, the model receives the dialogue so far, product information, and candidate intents. The grader computes per-turn accuracy and aggregated scores for model comparison.

## 4 BENCHMARK

**Task Formulation.** The evaluation task is defined as follows. The model input consists of three components: (1) the **dialogue history**, i.e., the full multi-turn bargaining context up to the current turn, ensuring that no prior information is omitted; (2) the **product information**, which includes real-world item descriptions, hierarchical category metadata with four levels, and listing prices; and (3) the **intent choice space**, a set of 20 candidate options randomly sampled from the complete intent space. The model output is a prediction of the buyer's true intent at each turn, selected from

the choice space. The central challenge is whether the model can continuously track and correctly identify buyer intent throughout multi-turn interactions.

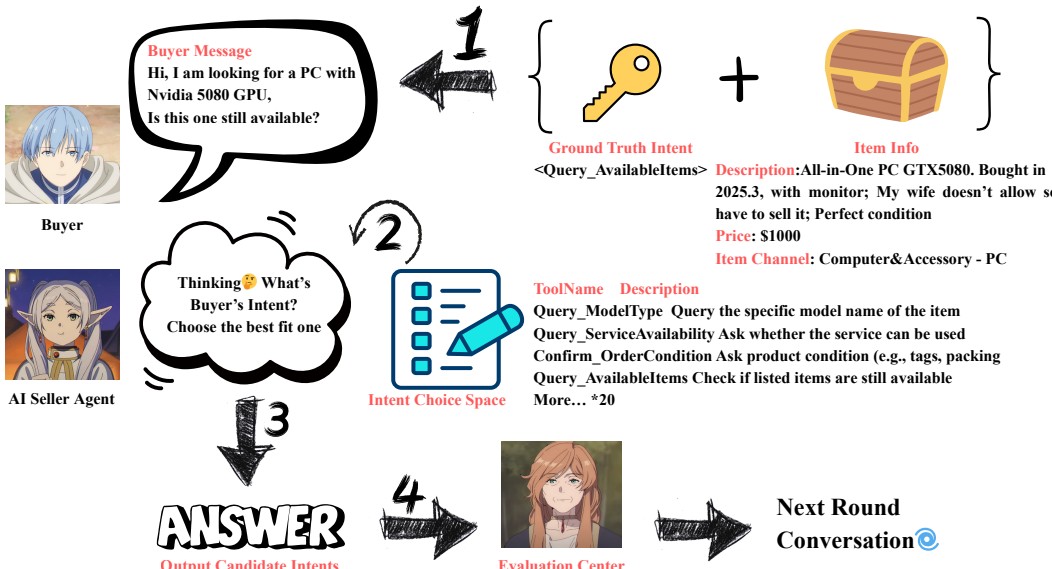

Figure 3: Workflow of intent recognition task. A buyer's message is paired with ground-truth intent and item information (Step 1). The AI seller agent infers the buyer's intent by selecting from a predefined intent choice space (Step 2). Candidate intents are produced as output (Step 3) and evaluated for correctness (Step 4), before proceeding to the next round of dialogue.

**Data Overview.** The benchmark dataset is constructed through the integrated pipeline of *Intent Factory*, *Problem Weaver*, and *Evaluation Center*. The resulting **Intent Space** is distilled from 10k authentic second-hand marketplace dialogues, yielding a three-level hierarchy of 17 intents, 39 actions, and 65 tools (Figure 5). On the listed product side, we curate 9,892 unique listings across 85 top-level categories, with metadata including title, description, price, and hierarchical category labels spanning four levels.

This coverage ensures diversity in both intent types and product domains, supporting realistic and scalable evaluation. The distribution of product categories and intents are shown in Figure 6, and additional dataset preparation details are provided in Appendix D.

| Statistic | Value |
|---|---|
| Total Items | 9,892 |
| Average Price | $209.03 |
| Unique Level 1 Categories | 85 |
| Unique Level 2 Categories | 700 |
| Unique Level 3 Categories | 1,336 |
| Unique Level 4 Categories | 1,611 |

Figure 4: Listed item statistics.

```
📋 Intent: Inquire_Product_Specification
   └ Action: Request_Product_Details
      └ Tool: API_QueryClothingLength
         └ Description: Query the specific length of clothing item for
buyer referen...
            └ Parameters: ['length_cm', 'fit_reference']

📋 Intent: Inquire_Shipping_Logistics
   └ Action: Check_Shipping_Policy
      └ Tool: API_QueryShippingPolicy
         └ Description: Query whether the item is eligible for free
shipping based o...
            └ Parameters: ['free_shipping', 'shipping_fee',
'shipping_notes']
```

Figure 5: Finalized three-level intent hierarchy: 17 intents, 39 actions, 65 tools.

**Metrics.** We categorize predicted intents into four types:

**Correct Intent (CI)** predictions that are in the choice space and exactly match the ground truth.

**Mismatched Intent (MMI)** predictions that in the choice space but doesn't match the ground truth.

**Missed Intent (MI)** intents that are in the ground truth but not predicted.

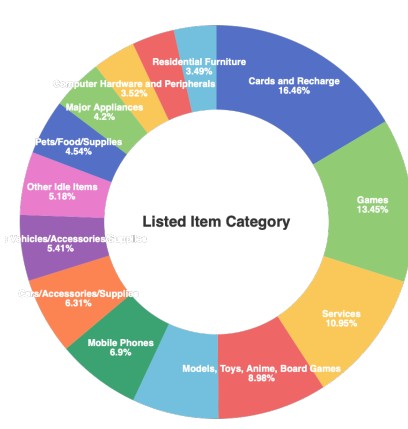
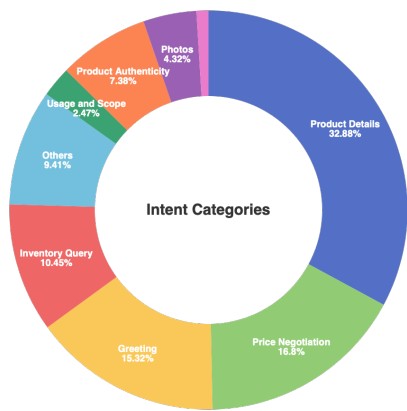

(a) Distribution of listed product categories.

(b) Distribution of intent categories.

Figure 6: Dataset composition: products vs. intents.

**Invalid Intent (II)** predictions outside the choice space.

These four categories capture different aspects of model behavior. To evaluate performance systematically, we define four complementary metrics that measure accuracy, coverage, and robustness:

1. **Intent-Precision (IP)**: the proportion of correct predictions among all predictions

$$IP = \frac{CI}{CI + MMI + II}$$

2. **Intent-Recall (IR)**: the proportion of ground-truth intents correctly predicted

$$IR = \frac{CI}{CI + MI}$$

3. **Intent-F1**: the harmonic mean of precision and recall, balancing the two aspects

$$F1 = \frac{2 \cdot IR \cdot IP}{IR + IP}$$

4. **Failure Rate (FR)**: the proportion of invalid predictions

$$FR = \frac{II}{CI + MI + II}$$

## 5 RESULTS

The main results are shown in Table 2.

| | Turn-2 | | | | Turn-3 | | | | Turn-4+ | | | |
|---|---|---|---|---|---|---|---|---|---|---|---|---|
| **Model** | **Precision** ↑ | **Recall** ↑ | **Failurs** ↓ | **F1** ↑ | **Precision** ↑ | **Recall** ↑ | **Failure** ↓ | **F1** ↑ | **Precision** ↑ | **Recall** ↑ | **Failure** ↓ | **F1** ↑ |
| gpt-5-chat-0807-global | **52.55** | 44.72 | **0.00** | 48.32 | **56.73** | 48.05 | **0.00** | 52.03 | **51.31** | 42.19 | **0.00** | 46.30 |
| gemini-2.5-pro-06-17 | 44.24 | 48.91 | 9.13 | 46.46 | 48.31 | 53.02 | 8.88 | 50.56 | 45.57 | **49.03** | 7.47 | 47.24 |
| gpt-41-0414-global | 50.08 | 46.43 | 0.17 | 48.19 | 51.94 | 47.08 | 0.11 | 49.39 | 48.85 | 44.35 | 0.08 | 46.49 |
| o3-0416-global | 47.09 | 42.70 | 0.86 | 44.79 | 50.55 | 44.83 | 0.11 | 47.52 | 47.69 | 41.44 | 0.43 | 44.35 |
| DeepSeek-V3-671B | 20.59 | 23.76 | 59.89 | 22.06 | 24.69 | 28.85 | 52.13 | 26.61 | 19.91 | 22.40 | 56.61 | 21.08 |
| kimi-k2 | 46.59 | 46.74 | **0.00** | 46.67 | 46.55 | 47.27 | **0.00** | 46.91 | 45.40 | 45.16 | 0.07 | 45.28 |
| qwen2.5-72b-instruct | 49.02 | **50.62** | 1.20 | **49.81** | 53.77 | **56.34** | 0.37 | **55.02** | 47.89 | 49.78 | 0.29 | **48.81** |
| qwen3-14b | 44.60 | 45.50 | **0.00** | 45.04 | 46.19 | 47.27 | 0.10 | 46.72 | 44.18 | 44.05 | **0.00** | 44.11 |
| qwen3-32b | 48.50 | 47.83 | 1.10 | 48.16 | 52.84 | 51.66 | 2.69 | 52.24 | 48.28 | 45.83 | 3.06 | 47.02 |

Table 2: Main experimental results. All metric values in this table are reported as percentages (%). Best and second results are in bold and underlined, respectively. Each column header "Turn-N" refers to tasks comprising N turns.

**Stable models show near-zero failure, while weaker ones collapse.** Failure directly reflects the reliability of a model acting as a seller agent. GPT-5-chat and Kimi-K2 achieve almost zero failure across all turns, while Qwen2.5-72B-Instruct and Qwen-14B remain similarly stable with values below 1.2%. In contrast, DeepSeek-V3-671B collapses with failure above 50%, and Gemini fluctuates near 8%, underscoring weaker robustness in multi-turn bargaining.

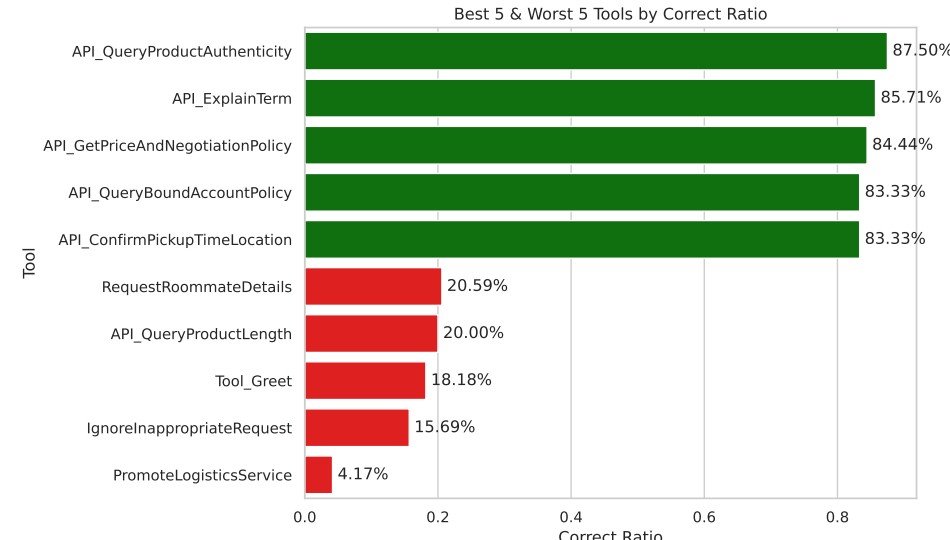

Figure 7: Accuracy distribution across tools, highlighting the best and worst performing categories.

**Additional turns improve understanding but amplify inconsistency.** Most models show small gains at Turn-3 and mild declines at Turn-4+. For example, Qwen2.5-72B-Instruct rises from 49.8% to 55.0% before dropping to 48.8%. This suggests additional turns can enhance task understanding, but longer dialogues mainly amplify inconsistency (precision loss) rather than coverage errors (recall remains steady).

**GPT-5 is the strongest performer, with Qwen competitive on F1.** GPT-5-chat combines the highest precision (56.7%) with perfect stability, yielding the most reliable overall performance. Qwen2.5-72B-Instruct achieves the best F1 balance (55.0% at Turn-3), supported by strong recall, while Qwen-32B is close behind. Kimi-K2 remains extremely stable but less precise, and DeepSeek-V3-671B performs worst with F1 below 27%.

**Comparison with Human Baseline.** To benchmark task difficulty, we conducted a small-scale human study. Three randomly selected non-expert users were asked to perform the evaluation on a sample of tasks. Their average precision was 73% and recall 65%, with consistent performance across tasks with different turn. This provides a practical reference point, showing that while LLMs lag behind expert-level reasoning, they are approaching general human performance on structured bargaining tasks.

**Precision distinguishes strong models, while recall remains steady.** Precision separates strong from weak systems more clearly than recall. GPT-5-chat leads with the highest precision, while Qwen2.5-72B-Instruct also performs well above 53%. Recall varies less across systems, with Qwen2.5-72B-Instruct consistently leading, followed by Gemini. Overall, recall remains steady while precision determines consistency of intent prediction.

**Structured queries are easy, while ambiguous or rare intents remain weak.** Accuracy varies sharply by intent type. Well-structured and explicit queries—such as product authenticity (INQUIRE_PRODUCT_AUTHENTICITY / QUERYPRODUCTAUTHENTICITY), terminology explanations (EXPLAINTERM), and policy lookups (GETPRICEANDNEGOTIATIONPOLICY)—reach

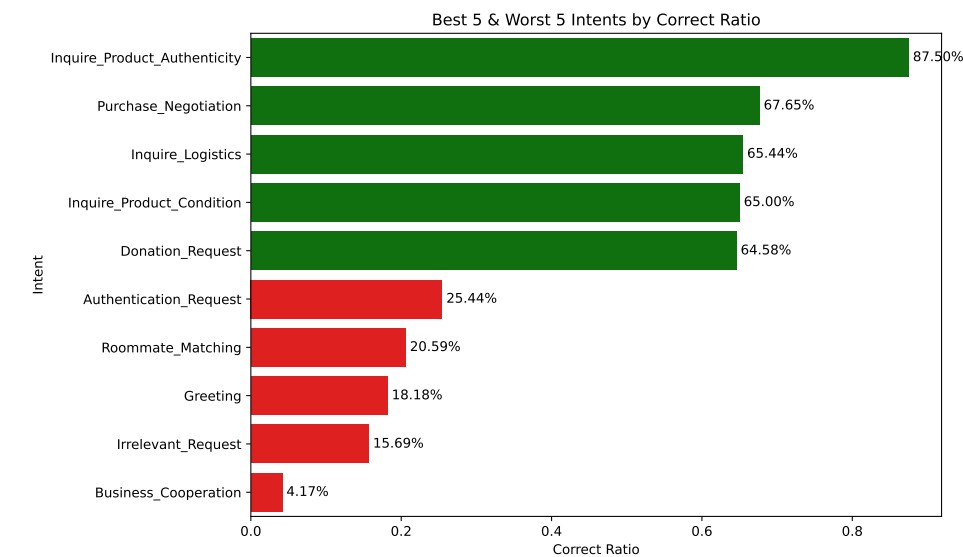

Figure 8: Performance aggregated at the intent level, showing precision variation across buyer intent categories.

83–87%. In contrast, ambiguous or infrequent intents perform poorly: PROMOTELOGISTICSSER-VICE (4.17%), IGNOREINAPPROPRIATEREQUEST (15.69%), TOOL_GREET (18.18%), and BUSI-NESS_COOPERATION (4.17%). These cases often span multiple domains or fall outside the core bargaining process, leading to weak coverage and frequent confusion. Taken together, results show models excel when intents are explicit but remain brittle under ambiguity, rarity, or cross-domain signals.

## 6 CONCLUSION

We introduced **BargainBench**, a large-scale benchmark for evaluating LLMs in multi-turn bargaining tasks within online second-hand marketplaces. Our framework integrates the *Intent Factory*, *Problem Weaver*, and *Evaluation Center*, enabling systematic generation and evaluation of negotiation dialogues with explicit ground-truth intents. Experiments show that strong models such as GPT-5 achieve stable performance with high precision, while others collapse under multi-turn settings. Structured and explicit intents are recognized reliably, while ambiguous or underrepresented intents remain difficult to capture.

Beyond e-commerce, the *intent–action–tool* hierarchy provides a general methodology for assessing social reasoning in domains such as diplomacy and colloborative game. By shifting from outcome-only scoring to turn-level intent tracking, BargainBench offers a process-grounded evaluation framework and a foundation for future research on intent understanding and negotiation capabilities in LLMs.

ETHICS STATEMENT

This work uses real-world e-commerce dialogues to construct the intent space in **Intent Factory**. Since these dialogues may contain sensitive personal information, the raw data cannot be released due to compliance requirements. To mitigate this, we verify that similar performance can be achieved with publicly available e-commerce dialogue datasets, which do not include personal identifiers. For the task synthesis stage in **Problem Weaver**, all product information is drawn from publicly accessible online listings. We believe the resulting benchmark poses minimal risks with respect to privacy, fairness, or safety.

This work does not involve human subjects or applications in high-stakes decision-making domains. The benchmark is designed for evaluating large language models in controlled experimental settings rather than deployment in sensitive or safety-critical environments. We therefore do not anticipate ethical concerns beyond the data anonymization and compliance considerations already described.

REPRODUCIBILITY STATEMENT

Code and scripts are provided in the supplementary material to replicate the empirical results.

The package implements all components of **BargainBench** (Intent Factory, Problem Weaver, Evaluation Center), with pipelines for dataset generation, model evaluation, and visualization. A ready-to-use evaluation script reproduces our main results with only an API key. Due to privacy constraints, we provide anonymized dialogue samples but include the full generation pipeline, enabling dataset regeneration. System requirements and parameters are documented to ensure faithful reproduction.

To ensure reproducibility, we release the full source code, configuration files, and documentation in the supplementary materials. The codebase implements all components of the **BargainBench** framework, including the **Intent Factory**, **Problem Weaver**, and **Evaluation Center**. The package contains pipelines for intent space generation, dialogue synthesis, and model evaluation, along with standardized interfaces for multiple LLMs. We also include visualization and analysis tools that allow researchers to inspect intermediate outputs such as the distribution of intents, the quality of scripted tasks, and model-specific evaluation results.

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

APPENDIX

## A  INTENT FACTORY: MULTI-AGENT PIPELINE & QUALITY METRICS

### A.1  FUNCTIONALITY OF EACH MODULE

- **Extractor** Baseline intent extractor that surfaces every candidate *intent–action–tool* triplet from raw marketplace dialogues and product descriptions without filtering or normalization.
- **Verifier** Gatekeeper that compares each newly extracted item against the current intent space; exact or near-duplicate entries are rejected, preventing redundancy.
- **Expert_guide** Domain-expert LLM invoked in a few-shot setting to re-label or re-categorize intents according to predefined taxonomic rules and canonical examples, ensuring semantic consistency across the hierarchy.
- **Maintainer** Post-processing aggregator that clusters semantically similar intents (via embedding similarity and synonym lists) and collapses redundant nodes, yielding a compact, non-redundant hierarchy while preserving coverage.

### GENERATION-QUALITY METRICS

**Coverage & Duplicate Ratio.**  We evaluate the mined hierarchy on a 10k marketplace dialogues:

**Variables**

- $G$ — number of ground-truth intents in the held-out dialogue set
- $M$ — number of intents our tools successfully match to at least one ground-truth intent
- $T$ — total number of intents we initially extract (before deduplication)
- $U$ — number of unique intents left after removing duplicates

**Formulas**

- Coverage $= \dfrac{M}{G}$
- Duplicate Ratio $= \dfrac{U}{T}$

In practice, Coverage > 95 %, so the smaller the Duplicate Ratio (i.e., the fewer unique intents we keep), the cleaner and higher-quality the final intent space.

**Refinement Curve.**  Figure 9a shows how each module progressively reduces the intent count. The 9 confirms that the final intent space size converges as raw data increases, indicating bounded growth and stable quality.

### A.2  PROMPTS

The prompt of extractor is shown in Prompt 1, and the prompt of verifier is shown in Prompt 2.

## B  PROBLEM WEAVER: DETAILED DEFINITION

### B.1  PROMPTS

The prompt of problem weaver is shown in Prompt 4

## C  DISCUSSION ON FRAMEWORK ADVANTAGES

Taken together, the **Intent Factory**, **Problem Weaver**, and **Evaluation Center** form an integrated pipeline for constructing and administering controlled bargaining evaluations. The bottom-up design

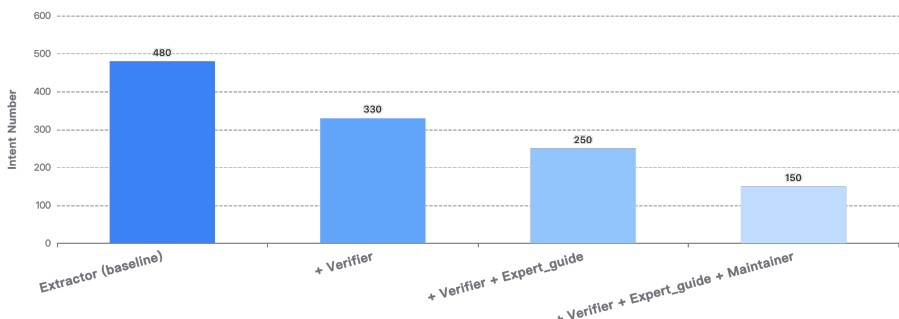

(a) Effect of progressively adding modules refining intent number. This show how we improve our Intent Factory Module. We would like to create a compact and concise intent space given sufficient intent coverage already. Basically, smaller the intent space size is, better the method is

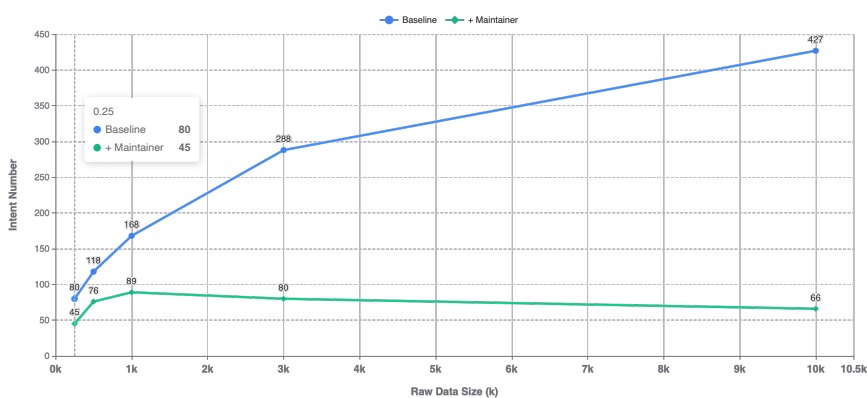

(b) Convergence of intent space size as raw data increases. Interestingly, green curve (+maintainer) which is our official conduct has dropped from 89 to 66 even input dialogue size multiply by 3x. LLM has found internal relation between these intents thus aggregate the cluster and remove the unnecessary ones

Figure 9: Results of the Intent Factory module.

ensures that every evaluation instance originates from realistic scenario data, is framed by a well-defined *intent–action–tool* hierarchy, and is paired with explicit turn-level ground truth.

Unlike benchmarks that stage end-to-end negotiation matches and judge performance by win–loss outcomes, our framework isolates the specific capability of *understanding* bargaining context: models are asked to infer buyer intent from dialogue history and structured choice spaces, rather than to simply generate plausible conversation turns. This design exploits the asymmetry between authoring and solving — models can readily produce convincing multi-turn interactions when given target intents, yet often fail to reliably recover those intents from completed exchanges.

By preserving verifiable ground truth at each turn, our method delivers interpretable, reproducible, and fine-grained performance measurements. Furthermore, because it is grounded in general principles of intent extraction, scenario synthesis, and structured evaluation, the approach can be directly adapted to other multi-turn, goal-oriented domains such as diplomatic negotiations, collaborative planning, and multi-party discussions.

# D    DATA PREPARATION DETAILS

The benchmark dataset is constructed through the integrated pipeline of *Intent Factory*, *Problem Weaver*, and *Evaluation Center*.

The **Intent Space** is derived from 10k authentic second-hand marketplace dialogues, focusing on extracting and aggregating buyer intents. We employ the advanced `qwen-plus-latest` model to perform large-scale extraction and refinement, consuming approximately 400M tokens; detailed prompting strategies are provided in the Appendix. The resulting structure contains 17 intents, 39 actions, and 65 tools (tools is the most granular level of intent hierarchical tree).

# E    EVALUATION TASK SAMPLE

There is a case of evaluation result in Figure 10 and Figure 11. Task generated by problem weaver, consist of system prompt, product info and context. Candidate model have to choose the best fit intent from a intent space of 20.

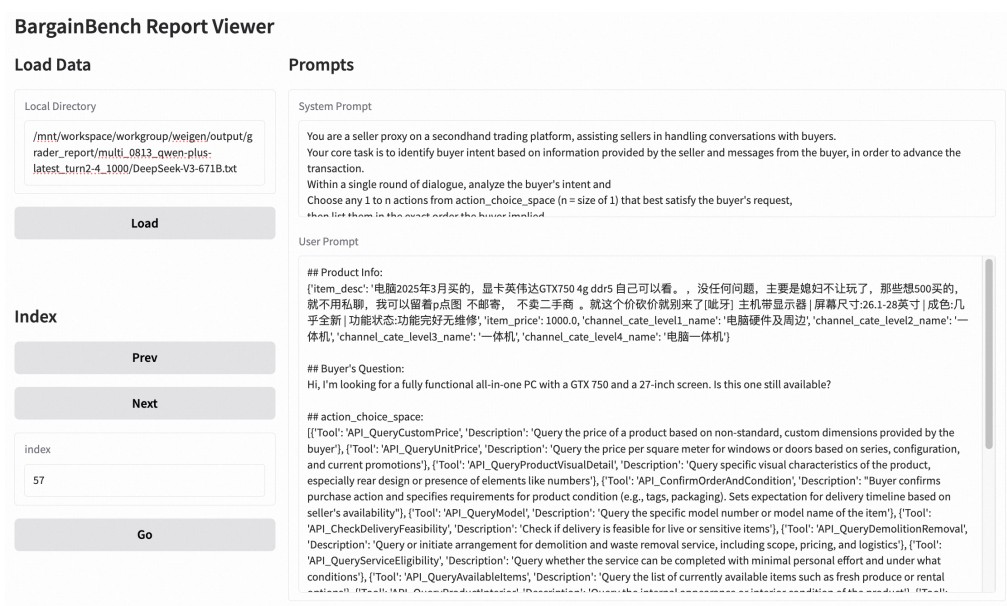

Figure 10: An illustration of evaluation sample. In User Prompt, the Chinese (original product description language) saying the PC was bought in 2025.3, with NVIDIA GTX750 GPU, in perfect condition and with monitor. He want to sell it because his wife doesn't allow it.

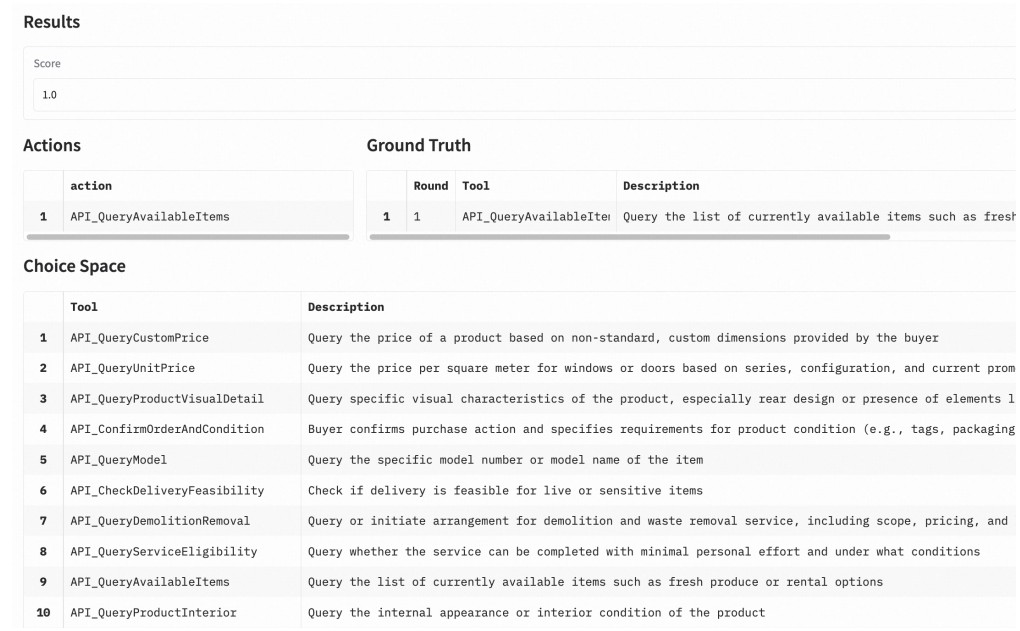

Figure 11: When Candidate Intents match with Ground Truth, LLM/candidate will get point on that task. Below are part of the Choice Space(20 in total ), where LLM has to choose the best one

## F  LLM USAGE

We used a large language model (ChatGPT/GPT-5) to assist with paper writing. Specifically, it was employed for improving grammar, clarity, and presentation of text, as well as for refining section structure and readability. All ideas, experiments, analyses, and conclusions are original to the authors, who remain fully responsible for the accuracy and integrity of the paper.

810

811 **Prompt 1: The Prompt of extractor**

812

813 You will use the Hierarchical Intent Decomposition (HID) framework to analyze the data. This
framework is used to extract and structure intents from dialogue data, forming a tree-like hierarchical
814 structure. HID decomposes intents into three orthogonal levels to ensure atomicity, orthogonality, and
815 scalability:

816 !!! Note, you are an advanced text understanding master and intent recognition expert with rich
knowledge. Please fully understand the text and provide intents, not limited to the examples I give.
817 - Intent (Root): The coarse-grained overall goal of the dialogue

818 - Action (Level 1 Branches): Mutually exclusive mid-level stages or categories, with no overlap. These
are orthogonal, and if needed, can be expanded through sub-branches.
819 - Tool (Leaves): Fine-grained atomic operations with a single intent and callable, with parameters (e.g.,
820 {'name': 'API_QueryPrice', 'description': 'Query item price', 'returns': {'price': {'type': 'number'}}}
821 ). Tools have no child nodes to maintain atomicity.

822 When processing input data (e.g., raw dialogue JSON with context, history, and features), follow these
steps:
823 1. Extract the root-level Intent based on the overall goal.

824 2. Decompose into orthogonal Actions, selecting from a predefined set or expanding if necessary
(ensuring no overlap).
825 3. Generate atomic Tools for each Action in JSON object format, including 'name', 'description', and
826 'returns' (including type/enum where appropriate).

827 4. For Tool generation, abstract and extract elements from the original text as much as possible, without
needing to be very specific.
828 Ensure the output is consistent, atomic (each Tool has a single purpose), orthogonal (no category
829 overlap), and extensible (if data introduces new intents, suggest new Actions/Tools without violating
830 rules). For the given input, generate the HID decomposition result.

831 I provide a dialogue segment, where "| buyer |" indicates the buyer's real person role, "| seller |" indicates
the seller's real person role, "| bot |" indicates the bot role. Please generate the HID decomposition
832 result for the "| buyer |" i.e., the buyer's role utterances in the dialogue content.

833 Output format: Please output in JSON format, including the keys, in English:
834 - Intent: string only

835 - Action: string only

836 - Tool: JSON format, including attributes like name,
description, returns, etc., name strictly requires English output, others no requirements
837

838

839

840

841

842 **Prompt 2: The Prompt of verifier**

843

844 You will be responsible for checking whether the newly added Intent-Action-Tool conflicts with the
existing action space. Assume that the new Intent-Action-Tool itself is valid (atomicity, clarity, etc., all
845 meet requirements), your only task is to check whether it conflicts with the Intent-Action-Tool in the
846 existing space, including category overlap (non-orthogonal) or functional similarity (duplication).

847 Each Intent-Action-Tool is a triplet, with "Intent", "Action", "Tool" three key values respectively.

848 In the action space, all Intent-Action-Tools are organized into three levels: Intent –> Action –> Tool
First, find the corresponding Tools information based on Intent and Action information, and check if
849 the target Tool conflicts with existing Tools.

850 Verification principles (focusing on conflict detection): - Each Tool is represented in JSON format,
851 including name, description, parameters, etc., need to compare each piece of information one by
one - Read and understand name, description information to judge whether there is duplication or
852 conflict - Check if the new Action overlaps with existing Actions (e.g., if existing has 'bargaining', new
853 'price negotiation' overlaps). - Check if the new Tool's function is similar to existing Tools, you can
854 assist judgment by checking description and parameters (e.g., if existing has 'API_QueryPrice', new
'API_GetItemCost' is functionally duplicate). - Ensure orthogonality: new items should not cross or
855 copy existing categories. - If there is no conflict at all, accept; otherwise reject.

856 Verification steps: 1. Compare the new Intent-Action-Tool with the existing space. 2. Output only one
857 of the following two (strictly follow the format, no additional explanation): - "1: No conflict, accept
858 new API" - "2: Has conflict, reject"

859 Input example: Existing space: 'Intent': 'Facilitate transaction', 'Actions': ['name': 'Information
860 query', 'Tools': ['name': 'API_QueryPrice']]; New item: 'Action': 'Bargaining', 'Tool': 'name':
861 'API_ProposeCounteroffer'. Output requirements: Please output in JSON format, including the
following keys: - status: 1 means no conflict, 2 means has conflict
862

863

**Prompt 3: intent-tool-action example**

"Inquire_Product_Details": { "Request_Specification": { "API_QueryProductSpec": { "description": "Query specific technical parameters of the product, such as power, voltage, or model specifications", "parameters": { "spec": { "type": "string", "description": "Technical specification requested by the buyer" } } } }, "Request_Visual_Info": { "API_QueryProductVisualDetail": { "description": "Query specific visual characteristics of the product, especially rear design or presence of elements like numbers", "parameters": { "has_number_on_back": { "type": "boolean", "description": "Indicates whether the back of the product has a number" }, "visual_description": { "type": "string", "description": "Textual description of the back appearance" } } } },

**Prompt 4: The Prompt of problem weaver**

Task description You are a master script-to-task writer. Your ONLY inputs are:
1. product_info - a short text containing the item description, price, and category. 2. ground_truth_action - an ordered list of API calls that the buyer must eventually issue.
Notes: Product info are match with format: item_desc, item_price, channel_cate_level1_name, channel_cate_level2_name, channel_cate_level3_name,channel_cate_level4_name
Your job is first generate buyer_question in English that naturally triggers the ground_truth_action. Feel free to add a plausible personal context so the question looks realistic.
Rules - Keep the question under 40 words. - Mention only the **first** API in the Ground Truth list; do not reveal the rest. - Translate any Chinese terms in product_info into natural English. - Do not quote the API names literally; phrase the concern in everyday language.
— Case
Sample Input product_info: Sam's Club Elsa Princess Dress, size 140. Worn once for photos—like new. ¥58. Kids' Apparel > Dresses > Princess Dresses. [API_CheckHeightFit, API_QueryShippingPolicy, API_CalculateOfferPrice]
Above action refers to: "API_CheckHeightFit": { "description": "Check if the product (e.g., bicycle) is physically suitable for the buyer based on their height or body measurements", "parameters": { "fit_result": { "type": "string", "enum": ["suitable", "too_small", "too_large", "uncertain"] }, "reason": { "type": "string" } } } },
"Inquire_Shipping_Logistics": { "Check_Shipping_Policy": { "API_QueryShippingPolicy": { "description": "Query whether the item is eligible for free shipping based on product details and seller settings", "parameters": { "free_shipping": { "type": "boolean", "description": "Indicates if the item is eligible for free shipping" }, "shipping_fee": { "type": "number", "description": "The shipping cost if not free, in yuan" }, "shipping_notes": { "type": "string", "description": "Additional notes about shipping" } } },
"API_CalculateOfferPrice": { "description": "Calculate a reasonable offer price based on item's marked price, bottom price, and negotiation stage", "parameters": { "offered_price": { "type": "number" }, "shipping_included": { "type": "boolean" } } },
Sample "buyer_question" output: "My daughter is 135 cm—will the size 140 be too big for her? Could you do 50 yuan with free shipping?"
— Inputs
product_info: product_info
ground_truth_action: ground_truth_action
—
Output Format
Your output should strictly follow the format. Otherwise, a cute kitty will starve for a dineer.
Ignore the parameters for now.
Please output in JSON format, including following keys: "buyer_question": string, a single-turn buyer question in English, in natural language.

