# OpenReview forum: "Evaluating Bargaining Skills in Online Second-Hand Marketplace with LLM Seller Agents"
_ICLR.cc/2026/Conference — ICLR 2026 Conference Withdrawn Submission_

### Official Review · Reviewer_eu3y · 2025-10-29

**Soundness:** 1
**Presentation:** 1
**Contribution:** 1
**Rating:** 0
**Confidence:** 4

**Summary:**

The paper argues that multi-turn bargaining is an important capability for large language models (LLMs). To measure LLMs bargaining capabilities when acting as “seller agents,” the authors propose a new framework that aims to isolate this capability by focusing on intermediate theory-of-mind actions rather than strictly assessing final outcomes. The framework is designed to measure LLMs ability to extract buyer intents. The authors create a benchmark using their framework based on real-world e-commerce bargaining dialogues. They then proceed to evaluate nine leading LLMs using this benchmark.

**Strengths:**

[**significance**] Interactive bargaining is an ecologically significant capability. Thus, developing benchmarks capable of isolating this capability in a scalable manner seems an important research direction.

**Weaknesses:**

[**quality**]
- Some claims miss supporting citations or discussion, e.g., lines 55 to 60 and lines 99 to 103.
- What is the source of the data? This is potentially an ethics violation and will be flagged as such. The “Ethics Statement” does not alleviate this concern (see questions).
- The paper misrepresents the prior work of [1]. Specifically, [1] explicitly measures ToM metrics in addition to final payoffs and proposes a flexible negotiation framework that allows for arbitrary issue constraints, e.g., [1] tests negotiations around rental agreements involving price and lease duration. Similarly, this can be used to describe arbitrarily complex e-commerce product constraints.


[**clarity**]
- Based on Section 3, it is unclear to this reviewer how the different components work. Section 3.1 describes the “Intent Factory”. This reviewer fails to understand what the “intent mining” operation is in reference to or how this works. In Section 3.2, the “Problem Weaver” is described, which seemingly uses outputs from Section 3.2 to generate dialogue turns – again: very unclear how this works. For example, how are multiple turns of buyer-side dialogue generated without seller-side responses?
- The usage of “Tool” is confusing and not coherent with literature in this space. Similarly, “Action” seems ill-chosen.
- Consistently wrong citation formatting using citet{} instead of citep{}, i.e., Authors et al. vs. (Authors et al.), makes this difficult to read.
- Overall formatting of the paper is difficult to follow. The figures used in this paper were not very informative to this reviewer in better understanding the core method and contributions. For example, Figure 2 and Figure 3 take up a significant amount of space, but are difficult to parse.


[**significance**] Frankly, it is hard to judge the significance of this work as the method and presented empirical results are very hard to follow for this reviewer. For a paper that primarily focuses on providing a novel benchmark, the results provided in Table 2 are a bit too light to support the strong claims of line 109-111 (“This [...] positions our work as a foundational contribution to the broader study of social intelligence evaluation [...]”).


[1] Davidson et al, Evaluating Language Model Agency through Negotiations, ICLR 2024

**Questions:**

Q1: Could you provide a worked example of the intents, actions, and tools?

Q2: What is the source of the data? What is the license and the underlying user consent?

Q3: What do you mean with “ground-truth intent” in this framework?

Q4: The paper repeatedly claims broad transferability of the presented framework to other problem settings. Could you provide more worked out examples supporting this claim?

**Details Of Ethics Concerns:**

The authors use a "10k authentic second-hand marketplace dialogues"(line 300), but do not discuss the source, license, or use consent conditions of the data. The provided ethics statement does not answer these questions either. The authors do claim to have verified that "similar performance can be achieved with publicly available e-commerce dialogue datasets" (l490-491), which makes this reviewer wonder why these results were not included.

---

### Official Review · Reviewer_aW2K · 2025-10-30

**Soundness:** 2
**Presentation:** 2
**Contribution:** 2
**Rating:** 2
**Confidence:** 3

**Summary:**

The paper proposes a pipeline for generating synthetic negotiation dialogues annotated with user intents, and a methodology for evaluating whether LLMs can accurately detect user intent.
The set of possible user intents is mined from real dialogues. Artificial dialogues are then synthesized with the help of an LLM.
Finally, various models are compared wrt their ability to detect intents at various turns of the conversation.

**Strengths:**

* The paper is motivated by an impactful use case for LLMs
* A significant effort went into crafting the framework and data
* The idea to leverage asymmetric task difficulty is nice
* The framework can be repurposed for other settings beyond bargaining

**Weaknesses:**

* The paper title and motivation promise more than what’s actually provided: “evaluating bargaining skills” is much broader than detecting user intent, but the paper is only about the latter.
* I found the description in Sec. 3 hard to follow.
* In Sec. 3.1 (“intent factory”), IIUC, ground truth annotations are extracted from real dialogue transcripts. But doesn’t this assume that LLMs can accurately extract that ground truth (including user intents) — which is precisely the ability that the paper wants to evaluate?
* It’s not clear what the role of “tools” is. Are these function calls? Or are they types of interactions that an agent can perform (such as “ask product description”, see Fig. 3)?
* A fundamental question is whether synthetic dialogue data is sufficient to measure the true abilities of LLMs “in the wild”. In this reviewer’s experience, synthetic data is much cleaner than real data, and is usually easier to handle for LLMs. Generating “dirty” synthetic data is truly difficult, and in my opinion and open challenge. Using synethic data for training models is fine, but evaluation ultimately needs to happen on real data. Since the paper’s very propose is to evaluate models, I question whether the data under consideration is appropriate to this end. This is my most serious concern about the paper.
* Overall, I find the results read somewhat like a “laundry list” of loosely connected findings. It’s difficult to see an emerging picture. What fundamental lessons do we learn about the abilities and shortcomings of today’s LLMs, and what conclusions can we draw about necessary future directions for the field?

**Questions:**

* How was the dialogue data collected?
* Line 418: “three randomly selected non-expert users”: from what population where these sampled? What does “random” mean here?
* Line 429: what is “accuracy” here? If I understand correctly, accuracy wasn’t used before in the paper

---

### Official Review · Reviewer_ZXgG · 2025-10-30

**Soundness:** 1
**Presentation:** 1
**Contribution:** 2
**Rating:** 2
**Confidence:** 4

**Summary:**

The paper collects a dataset of product descriptions and dialogues. It then uses these dialogues to extract intents across different turns. These intents form a set of Intent Space. Next, scripts for the negotiation dialogs are generated, given a sample of intents and product descriptions. Finally, these turns are fed to an LLM that should map the intent of each turn to a ground-truth output out of choices of intents.

**Strengths:**

- The paper builds the setup on real-world tasks and descriptions that are drawn from real-world data.

**Weaknesses:**

- The paper is very unclear and hard to follow. There are so many terms that were introduced and never explained. In Table 1, what does scalability mean? In Figure 4, what are levels 1/2/3? What is domain knowledge? How can this be instantiated in this setup?

- The paper mentions it is a benchmark for negotiation or bargaining, but it seems the benchmark is measuring how models can identify the intent of different turns in logs of fixed, pre-generated conversations. This becomes clear only in the middle of the paper. It is also confusing to compare to previous work, which designed negotiation benchmarks for LLMs where agents themselves negotiate rather than LLMs generated scripted logs according to ground truths of intent that should be present in each turn.

- It is not clear whether models that identify intent from conversations can also identify these intents (and use them) when actually engaging in negotiation. Ultimately, the paper does not measure end-to-end tasks or the performance of LLMs in negotiation, and the current results may not be directly applicable.

- Given that, the title of the paper is misleading because there is no "seller agent" (i.e., an LLM that actually responds to

- The paper motivates the setup with intent drift, but it is not clear how significantly the drift is happening across these conversations or how to define drift. Figure 3 shows that the intent choice space can be quite direct/straightforward, such as asking about availability or conditions. These are quite obvious from the text of the turns. They don't involve interpreting high-level cues such as willingness to buy items, etc.

- The paper mentions that one of the motivations is that previous work on negotiation only evaluates the final outcome, which is not very accurate. For example, please refer to [1,2], which are also highly related to ToM in negotiation.

- There are not enough details about the human baseline (e.g., how many users, how many samples).

Overall, due to significant clarity issues of the paper and the limited setup of an agentic application, I don't believe the paper is ready for acceptance.

Minor:
- Figure 1 is overlapping with the text
- Figure 4 should be table
- The paper mixes \citet with \citep, and this can be hard to read. citet should be used when referring to authors instead of repeating the citation (e.g., "Xia et al. Xia et al. (2024)").
-  Figure 2 is never referenced. Is this the actual setup of the paper?

[1] https://proceedings.neurips.cc/paper_files/paper/2024/hash/984dd3db213db2d1454a163b65b84d08-Abstract-Datasets_and_Benchmarks_Track.html
[2] Playing repeated games with large language models (https://www.nature.com/articles/s41562-025-02172-y)

**Questions:**

- Where are the dialogues, descriptions, etc. coming from?
- Since the agents are not negotiating, what is the importance of having tools?
- How is the dataset released, given that raw dialogues can't be released? What are the mechanisms to ensure that descriptions and intents don't have PII?

**Details Of Ethics Concerns:**

The paper mentions 10k dialogues and product descriptions but I didn't find any details about how these were collected. Are these scraped from second-hand marketplaces (e.g., eBay)? Does this adhere to the rules of these websites? How did the authors ensure no PII exist?

---

### Official Review · Reviewer_wRA9 · 2025-11-02

**Soundness:** 2
**Presentation:** 2
**Contribution:** 2
**Rating:** 2
**Confidence:** 4

**Summary:**

Study process-based over outcome-based negotiations. Design a pipeline for taking raw marketplace conversations, mining intents and then evaluating how well a language model is at selecting tools and answering specific questions for a given intent. The authors find that the SOTA models perform well on the evaluation and are able to correctly identify intents. They release a benchmark called BargainBench.

**Strengths:**

The paper is tackling an important area. As AI agents become increasingly used, they will begin to represent individuals in the wild, and having good evaluations for negotiations is important.
This is one of the few papers that explicitly designs an agentic system for evaluating negotiation performance.

**Weaknesses:**

Past works on negotiations also have process-level evaluations rendering the claim “Most existing benchmarks assess bargaining only by final outcomes such as success rate or payoff, without capturing how agents reason across turns…” false. Example, in Davidson et al. they study faithfulness and rule following as process-level evaluations.

I feel like the contributions of this work are a little overblown. At the end of the day this is another benchmark for evaluating negotiation performance. In fact, I think the more interesting contribution is the underlying dataset of real negotiations in the wild, which wasn’t presented in sufficient detail to my liking. I realize that the data is under compliance review, but I would have liked for it to have passed compliance review before being submitted.

The fact that good models have near-zero failure leads me to question the importance of the benchmark if it’s already saturated.

Overall, I feel like the contribution of this paper isn't worthy of publication.

Minor points:
- citation type is off throughout paper [37-38,129],
- figure 1 overlaps with text
- figure 7 is never mentioned in the text, I don’t know what model its capturing.
- The names intent factory, problem weaver, and evaluation center are unnecessary, I would prefer simpler naming that just includes what they are doing

**Questions:**

Can you provide any more context on where the data came from?

---

### Note · Authors · 2025-12-01

I have read and agree with the venue's withdrawal policy on behalf of myself and my co-authors.